# Salicylic Acid and Jasmonic Acid Increase the Polysaccharide Production of *Nostoc flagelliforme* via the Regulation of the Intracellular NO Level

**DOI:** 10.3390/foods12050915

**Published:** 2023-02-21

**Authors:** Cheng-Feng Han, Shu-Ting Liu, Rong-Rong Yan, Jian Li, Ni Chen, Le-Le Zhang, Shi-Ru Jia, Pei-Pei Han

**Affiliations:** 1State Key Laboratory of Food Nutrition and Safety, Tianjin University of Science and Technology, Tianjin 300457, China; 2Key Laboratory of Industrial Fermentation Microbiology, Ministry of Education, College of Biotechnology, Tianjin University of Science and Technology, Tianjin 300457, China

**Keywords:** salicylic acid, jasmonic acid, *Nostoc flagelliforme* polysaccharides, nitric oxide levels, physicochemical properties

## Abstract

To significantly improve the polysaccharide production of *Nostoc flagelliforme*, a total of 12 chemicals were evaluated for their effects on polysaccharide accumulation. The results showed that salicylic acid and jasmonic acid increased the accumulation of the polysaccharides in *N. flagelliforme* significantly, by more than 20%. Three polysaccharides, namely control-capsule polysaccharide, salicylic acid-capsule polysaccharide, and jasmonic acid-capsule polysaccharide, were extracted and purified from *N. flagelliforme* under normal, salicylic acid, and jasmonic acid culture conditions, respectively. Their chemical compositions slightly differed regarding the total sugar and uronic acid contents, with average molecular weights of 2.06 × 10^3^, 2.16 × 10^3^ and 2.04 × 10^3^ kDa, respectively. They presented similar Fourier transform infrared spectra and no significant difference in antioxidant activity. It was revealed that the salicylic acid and jasmonic acid significantly increased the level of nitric oxide. By investigating the effects of the exogenous nitric oxide scavenger and nitric oxide donor on the nitric oxide levels and polysaccharide yield of *N. flagelliforme*, the results showed that the increase in intracellular nitric oxide levels might be an important factor promoting the accumulation of polysaccharides. These findings provide a theoretical foundation for enhancing the yield of secondary metabolites by regulating the intracellular nitric oxide levels.

## 1. Introduction

Because of the diverse effects of microalgal polysaccharides, such as antioxidative, antiviral, antitumor, and other biological activities, there has been increasingly more in-depth research in this area [1,2]. *Nostoc flagelliforme* is a terrestrial cyanobacterium with tremendous economic value that is used as a vegetable in Chinese cuisine [3,4]. It is distributed in arid and semi-arid areas and can secrete abundant extracellular polysaccharides to protect itself from drought stress [5,6,7]. The capsule polysaccharide (CPS) bound to the cell surface and the extracellular polysaccharide (EPS) released to the surrounding environment are the two main forms of the *N. flagelliforme* extracellular polysaccharides [8,9]. The yield of CPS was higher than that of the EPS, but had a similar performance to EPS [1]. Therefore, CPS was the key analysis object in this study. Many studies have aimed to increase the yield of polysaccharides in *N. flagelliforme*, especially the accumulation of polysaccharides under various stress conditions, such as nutritional stress, salinity, suboptimal light intensity, and light quality [1,10,11,12]. The results have shown that the accumulation of polysaccharides in *N. flagelliforme* is induced when the cells are exposed to various stress factors.

According to previous studies, various stress factors can improve the accumulation of polysaccharides but also often change their structure and properties. In contrast, chemicals that act as metabolic elicitors or inducers can directly regulate cellular metabolism to promote the synthesis and accumulation of *N. flagelliforme* polysaccharides, without altering their structure and properties [13]. Recent studies have shown that polysaccharide accumulation could be enhanced by salicylic acid (SA) and jasmonic acid (JA), and the enhancing effects of these chemical inducers were concentration-dependent [14,15].

Nitric oxide (NO) has emerged as a key signaling molecule that exerts various signaling functions in many biological functions of plants [16]. In recent years, it has been reported that NO participates in the accumulation of the plant secondary metabolites induced by biotic or abiotic stresses. Polysaccharide is one of the secondary metabolites with extensive physiological functions, and its relationship with NO has attracted attention in recent years. Wang et al. found that the addition of exogenous NO donors could significantly increase the production of the extracellular and intracellular polysaccharides of *Ganoderma lucidum* in submerged fermentation, but the physiological mechanism through which NO induced the *Ganoderma lucidum* polysaccharides has not been clarified [17]. It was found in the protocorm of *Dendrobium candidum* that low temperature induction could open the NO signal pathway and promote the accumulation of polysaccharides [18]. It has been reported that exogenous SNP could increase the content of NO in cells, and NO could induce the production of vinblastine in catharanthus roseus cells [19]. These observations suggest the existence of a NO-mediated signaling pathway that mediates the elicitor-induced biosynthesis of secondary metabolites in plant cells.

SA and JA are signal molecules that can stimulate the biosynthesis of plant secondary metabolites through different signal transduction pathways. Research has shown that NO and JA are involved in elicitor-induced hypericin biosynthesis [20]. Crosstalk between SA and NO plays a role in the upregulation of the antioxidant capacity and the photosynthetic adaptability of plants in response to environmental assaults such as drought and salinity [21]. Here, we screened SA and JA among 12 chemical inducers that can significantly improve the polysaccharide production of *N. flagelliforme* cells. These two inducers not only improved the yield of polysaccharides but also maintained their original structure and function. In order to explore the possible mechanism of SA and JA inducing *N. flagelliforme* polysaccharide synthesis and provide a new idea for the accumulation of *N. flagelliforme* polysaccharide, we measured the content of NO in *N. flagelliforme* cells induced by SA and JA, and found that NO played a key role in the process of *N. flagelliforme* polysaccharide synthesis.

## 2. Materials and Methods

### 2.1. Materials and Reagents

The *N. flagelliforme* cells (TCCC11757) used in this experiment were provided by the Tianjin Key Lab of Industrial Microbiology (Tianjin, China).

The standard monosaccharides were purchased from Sigma Chemical Co. (St. Louis, MO, USA). H_2_O_2_, MgCl_2_, CaCl_2_, methylene blue (MB) and 2,2′-azo-bis(2-amidinopropane)-dihydrochloride (AAPH), propyl gallate (PG), 2,4-dichloro phenoxy acetic acid (2,4-D) and gibberellic acid (GA), salicylic acid (SA) and jasmonic acid (JA) were purchased from (Solarbio, Beijing, China). All of the chemicals used were analytically pure. PG and abscisic acid (ABA) were made in amber glass vials and stored in the dark at −20 °C. The remaining reagents were prepared in amber glass bottles and stored at 4 °C away from light.

### 2.2. Strains and Growth Conditions

The culture conditions of *N. flagelliforme* were 25 °C, 60 μmol photons m^−2^ s^−1^, and BGII medium. They were then inoculated (10%, *v*/*v*) to Erlenmeyer flasks after 10 days of continuous culture. At the early stage, the cells were cultivated in 100 mL Erlenmeyer flasks each containing 50 mL of BG-11 medium. In the early stage, the cells were grown in 2000 mL Erlenmeyer flasks each containing 1500 mL of BG-11 medium. The flask was shaken manually twice a day for 15 days (approaching stationary phase on day 15) [12].

### 2.3. Extraction and Purification of the CPS

In this paper, the crude polysaccharides were extracted on the base of reference literature [1].

The cell culture solution was centrifuged at 4000 r for 10 min, and the cells were collected, frozen, and dried. The freeze-dried cells were added to distilled water and then bathed in water at 80 °C for 4 h. The supernatant was collected by centrifugation. The content of capsular polysaccharide in the supernatant could be determined using the phenol-sulfuric acid method.

After the supernatant was concentrated by ultrafiltration, 80% ethanol was added to precipitate it, and it was placed at 4 °C overnight, centrifuged, collected the precipitate, and freeze-dried to obtain the capsule polysaccharide powder. The capsule polysaccharide powder was prepared into polysaccharide aqueous solution, separated by DEAE-52 column chromatography, purified by Sephadex G-100 column chromatography, and the impurities were removed. The collected sugar-containing liquid was freeze-dried to obtain the purified capsule polysaccharide of *N. flagelliforme*, and stored at −80 °C for standby.

### 2.4. Yield of Crude Polysaccharide

The total polysaccharides were detected using a phenol-sulfuric acid assay.
The yield of crude polysaccharide (%) = W_1_/W_2_ × 100%(1)

W_1_: the content of polysaccharides

W_2_: the weight of *N. flagelliforme* powder

### 2.5. Chemical Analysis

The total sugar was determined according to the phenol-sulfuric acid method. The total protein content was determined by Bradford’s method using bovine serum albumin as the standard [22]. For determination of the uronic acid content, please refer to the method of Fan Liuping et al. [23]. The total polyphenol contents were determined using Folin–Ciocalteu’s reagent according to the national standard method of the People’s Republic of China [24]. The monosaccharide compositions of the polysaccharides were analyzed with reference to previous reports [25]. The homogeneity and molecular weight (MW) of the polysaccharides were analyzed according to the method previously reported [12].

### 2.6. Structural Characterization

#### 2.6.1. Ultraviolet (UV) Spectra

In the wavelength range of 200–400 nm, 1 mg/mL polysaccharide solution was taken and scanned by UV −2401 ultraviolet spectrophotometer of SHIMADZU Corporation (Kyoto, Japan), to obtain the ultraviolet absorption spectrum.

#### 2.6.2. Fourier-Transform Infrared (FT-IR) Spectra

Polysaccharide samples of 1 mg and KBr of 100–200 mg were mixed, ground and pressed in an agate mortar, and analyzed at wave numbers of 4000 to 400 cm using a VECTOR 22 Fourier transform infrared spectrometer (Thermo Nicolet Corporation, America).

### 2.7. Evaluation of Antioxidant Activity

First, 3 mL of polysaccharide solution with different concentrations (0.5, 1.0, 1.5, 2.0 and 2.5 mg/mL) and 1 mL of DPPH-ethanol solution (0.1 mmol/L) were mixed evenly, and the absorbance at 517 nm was determined after 30 min in dark. Vitamin C was used as the positive control. The DPPH radical scavenging activity was calculated as the scavenging rate (%) using the equation [12].
DPPH radical scavenging activity (%) = [1 − (A_1_ − A_2_)/A_0_] × 100%(2)

A_0_: the absorbance of the control (without sample)

A_1_: the absorbance in the presence of the sample and DPPH

A_2_: the absorbance of the sample blank (without DPPH)

The determination of the hydroxyl radical scavenging activity was modified according to the method in the reference [22]. The reaction mixture consisted of 1 mL of polysaccharides of varying concentrations, 1 mL of FeSO_4_ (5 mmol/L), 1 mL of salicylic acid ethanolic solution (5 mmol/L), and 1 mL of H_2_O_2_ (5 mmol/L). The final mixture was incubated for 30 min at 37 °C and the absorbance of the mixture was measured at 510 nm. The hydroxyl radical scavenging activity was calculated as a scavenging rate (%) using the equation:OH radical scavenging activity (%) = [1 − (A_1_−A_2_)/A_0_] × 100%(3)

A_0_: the absorbance of the control (without sample)

A_1_: the absorbance in the presence of the sample and hydroxyl radical

A_2_: the absorbance of the sample blank (without hydroxyl radical)

Then, 10 mL of ABTS solution and 10 mL of potassium persulfate solution (2.6 mmol/L) were mixed evenly and reacted in the dark for 6 h. PBS was then diluted as buffer to make its absorbance about 0.7 at 734 nm, and the ABTS reaction solution was prepared. Then, 0.5 mL of polysaccharide solution with different concentrations (0.5, 1.0, 1.5, 2.0, and 2.5 mg/mL) was mixed with 3 mL of ABTS reaction solution, and the absorbance at 734 nm was determined after standing for 10 min. Vitamin C was used as the positive control. The ABTS^+^ free radical scavenging rate was calculated as follows [12]:ABTS^+^ radical scavenging activity (%) = [1 − (A_1_ − A_2_)/A_0_] × 100%(4)

A_0_: the absorbance of the control (without sample)

A_1_: the absorbance in the presence of the sample and ABTS^+^

A_2_: the absorbance of the sample blank (without ABTS^+^).

### 2.8. Determination of Nitric Oxide Content

The content of nitric oxide was determined using the DAF-FM DA kit.

First, 1 mL of *N. flagelliforme* cell suspension was centrifuged at 4 °C for 10 min at 4000 rpm. Then, we discarded the supernatant and added 200 μL 5 μM DAF-FM DA and incubated it at 37 °C in a shaker for 40 min, and then washed it three times with PBS. Finally, the cells were resuspended with 200 μL PBS, and the fluorescence intensity of each experimental group was measured at 495 nm excitation wavelength and 515 nm emission wavelength [26].

### 2.9. Statistical Analysis

The data were analyzed by analysis of variance (ANOVA) or independent-samples t-test using the SPSS statistical software (version 20.0). All of the samples were run in triplicate. The significance level was set at *p* < 0.05.

## 3. Results and Discussion

### 3.1. The Screening of Chemical Inducers for Increasing the Polysaccharide Accumulation

Research based on the effect of previous different chemicals on the accumulation of target products of different microalgae [27] resulted in, 12 chemicals from six groups, including antioxidants, oxidants, signal transducers, metal ions, auxins, and vitamins, being selected for the evaluation of their effects on the polysaccharide accumulation in *N. flagelliforme*.

Table 1 shows the effects of the chemical inducers on the polysaccharide production in *N. flagelliforme* cells, which can be classified into two groups. In the first group, the polysaccharides production of cells treated with 2,4-D, GA, Ca^2+^, MB, H_2_O_2_, PG, JA, SA, and N-acylated homoserine lactones (AHLs) increased at a certain concentration compared with their corresponding control groups. In the second group, the chemical inducers induced no significant changes in the polysaccharides of the *N. flagelliforme* cells.

Under the effects of SA and JA at a concentration of 1 mg/L, the polysaccharide yield of *N. flagelliforme* increased by 35.33% and 58.57%, respectively. As plant hormones, SA and JA are naturally present in many plant cells and participate in a variety of regulatory pathways. Studies have shown that JA significantly promoted the synthesis of oil in *Chlorella* [28], methyl jasmonate (MeJA) significantly increased the content of polysaccharides in Lentinus edodes, and SA induced the synthesis of Ganoderma lucidum polysaccharides [29]. There is evidence that they both constitute components of plant defense response systems as signal molecules. Moreover, studies have shown that they can participate in the oxidative defense mechanism of algae against environmental stress. Therefore, the effects of SA and JA on the physicochemical properties and antioxidant activities of *N. flagelliforme* polysaccharides were further investigated.

### 3.2. The Effects of SA and JA on the Culture of N. flagelliforme

As shown in Figure 1A, the biomass of *N. flagelliforme* was not significantly affected by the culture conditions. The electron yield of the *N. flagelliforme* cells and the activity of photosystem II was not significantly affected. These results indicated that these two chemical inducers had no toxic side effects (Appendix A).

### 3.3. The Effects of SA and JA on the Physicochemical Properties and Antioxidant Activity of Polysaccharides

#### 3.3.1. Analysis of the Basic Physicochemical Properties

Chemical characteristics are important indicators for identifying natural polysaccharides [30]. Their main chemical components included the total sugar, protein, uronic acid, and total phenol concentrations, which are shown in Table 2. JA-CPS contained the highest total sugar (89.59%), indicating its high purity. SA-CPS contained the highest amount of protein (0.74%), while control-CPS contained the highest amount of uronic acid (15.03%). The molecular weights of the control-CPS, SA-CPS, and JA-CPS were 2.06 × 10^3^ kDa, 2.16 × 10^3^ kDa and 2.04 × 10^3^ kDa, respectively.

Through the analysis of the monosaccharide composition, the conclusion was consistent with the previous studies on *N. flagelliforme* polysaccharides [12]. All three polysaccharide samples were heteropolysaccharides, composed of aldohexose (glucose, galactose, and mannose), ketohexose (fructose), pentose (ribose, xylose, and arabinose), and deoxysugar (rhamnose and fucose) plus one uronic acid (glucuronic acid). As shown in Table 2, in all the experimental groups, glucose was the most abundant polysaccharide in CPS, followed by mannose and galactose. Except for glucose, there were no significant differences in the monosaccharide composition between the samples.

#### 3.3.2. UV and FT-IR Analysis

As shown in Figure 2A, the absorbance decreased monotonically in the wavelength range of 220–380 nm, with no characteristic absorption peaks of nucleic acids (260 nm) and proteins (280 nm), indicating that there were no or very few nucleic acids and proteins in the samples. The above results were consistent with the analysis in Table 2.

The infrared spectra of the control-CPS, SA-CPS, and JA-CPS are shown in Figure 2B and were highly similar. The characteristic absorption of hydroxyl made the infrared spectrum of the *N. flagelliforme* polysaccharides show a broad and intense peak near 3416 cm^−1^. The signal at 2926 cm^−1^ was attributed to the stretch vibration of the C–H bond [31]. In addition, the stretching peak of 1636 cm^−1^ and the weak stretching peak of 1412 cm^−1^ could indicate the presence of uronic acid in the polysaccharide structure, because these two peaks are characteristic signals of deprotonated carboxyl groups [32]. The deformed vibration of the C–H bond could be analyzed in the peak observed in the range 1400–1200 cm^−1^. The presence of furanose ring in polysaccharides is generally in the range of 1100–1010 cm^−1^. It can be seen from Figure 2B that the *N. flagelliforme* polysaccharide had two strong absorption peaks in the range of 1100–1010 cm^−1^, so it was speculated that there was a furanose ring [24]. Some studies have shown that the FTIR bands of galactose and glucose were the strongest at 1078 cm^−1^ and 1035 cm^−1^ respectively [33]. Our results show that the control-CPS, SA-CPS, and JA-CPS had a strong absorption peak at 1038 cm^−1^, indicating that they were rich in glucose, which was consistent with the monosaccharide composition analysis (Table 1). The vibration absorption peak at 1200 and 900 cm^−1^ was the C-O-C bond stretching vibration peak of the pyran ring in the control-CPS, SA-CPS, and JA-CPS [1].

#### 3.3.3. In Vitro Antioxidant Activity

DPPH is an accurate, convenient, and quick method used to analyze the free radical scavenging ability of natural compounds [34,35]. The principle of this method is that the DPPH-H adduct in the form of free radicals is formed through certain reactions in the presence of hydrogen supplying antioxidants [35]. Polysaccharides can form stable DPPH compounds in a dose-dependent manner. The polysaccharides showed a clear increase in the antioxidant activity as the concentration increased from 1 to 5 mg/mL (Figure 3A). The highest scavenging activity was observed at 5 mg/mL, and it was higher in the JA-CPS (54.29 ± 6.47%) than in the control-CPS (42.27 ± 3.89%) or SA-CPS (42.66 ± 5.91%).

The hydroxyl radical (·OH) is a free radical with strong oxidation ability, which easily oxidizes various organic and inorganic substances. It has a high oxidation efficiency and fast reaction rate and is an important reactive oxygen species that causes lipid peroxidation, nucleic acid breakage, and protein and polysaccharide decomposition in biological tissues. Compounds with hydroxyl radical scavenging ability can generally chelate iron ions and make them inactive in the Fenton reaction [36]. The scavenging abilities of the control-CPS, SA-CPS, and JA-CPS against the hydroxyl radical are shown in Figure 3B. The scavenging activity of the control CPS, SA-CPS, and JA-CPS was positively correlated with the polysaccharide concentration (<5 mg/mL). SA-CPS and JA-CPS exhibited a similar scavenging ability, which was slightly lower than that of the control-CPS.

The determination of the ABTS^+^ radical is generally used to estimate the total antioxidant capacity of different compounds. The free radical scavenging ability of ABTS^+^ of CPS, SA-CPS, and JA-CPS is shown in Figure 3. When the concentration of *N. flagelliforme* polysaccharides increased from 0 to 0.5 mg/mL, their ABTS^+^ radical scavenging activity dramatically increased, followed by a gradual further increase from 0.5 to 2.5 mg/mL. At a concentration of 2.5 mg/mL, the ABTS^+^ radical scavenging rates of the control-CPS, SA-CPS and JA-CPS reached 99.59%, 99.95%, and 99.21%, respectively. These results indicated that all of the polysaccharide samples had a good radical scavenging effect in the ABTS^+^ assay. Transforming reactive free radicals into a stable state, while providing electrons to free radicals to achieve the chain reaction of terminating free radicals, this process has been proved to improve the scavenging capacity of ABTS^+^ free radicals [37,38].

### 3.4. The Possible Mechanisms through Which SA and JA Affect the Accumulation of Polysaccharides in N. flagelliforme

Studies have shown that SA and JA can induce changes in the intracellular reactive oxygen species (ROS) and NO levels [39]. Therefore, the contents of the NO, ROS, and malondialdehyde (MDA) were determined in this study. As shown in Appendix A, the contents of the ROS and MDA in the *N. flagelliforme* cells in the experimental group were significantly lower than those in the control group, indicating that SA and JA inhibited the production of ROS and MDA and broke the redox balance of *N. flagelliforme* cells. Based on the above phenomena, this study measured the activities of the superoxide dismutase (SOD) and catalase (CAT) in the *N. flagelliforme* cells under three culture conditions, as shown in Appendix A. The results show that the antioxidant defense level of the *N. flagelliforme* cells decreased under SA and JA stimulation, which may be due to their effect on the intracellular ROS level. 

Reports have demonstrated that NO could scavenge ROS and protect plant cells from damage. Another study reported that JA could induce *Sophora flavescens* suspension cells to produce NO by increasing the activity of nitric oxide synthase and finally promote the accumulation of matrine [40]. In addition, in plants, NO and SA can promote the accumulation of each other [41]. Therefore, it is speculated that SA and JA could affect polysaccharide production by regulating the level of NO. The following studies monitored the changes in the polysaccharide content by varying the amount of NO levels.

#### 3.4.1. Changes in the NO Content under SA and JA Induction

As a signaling molecule, NO production can be induced by a variety of plant hormones, such as cytokinin [42], auxin [43] and SA [44]. Furthermore, NO usually interacts with these hormones to regulate different physiological processes [45]. Here, the effects of 1 mg/L of SA and JA on the NO levels were investigated. As shown in Figure 4, the NO content in the cells of *N. flagelliforme* in the experimental group was significantly higher than in the control group.

To investigate the key regulatory role of NO in the promotion of polysaccharide synthesis, the influence of C-PTIO on the biomass, NO content, and polysaccharide production were analyzed by adding the exogenous NO scavenger C-PTIO to cultures after 6 days of SA or JA induction.

#### 3.4.2. The Effect of the Exogenous NO Scavenger C-PTIO on the NO Level and Polysaccharide Production

As shown in Appendix A, there was no significant change in the biomass of the experimental group with the C-PTIO added on the sixth day, compared with the control group without C-PTIO, indicating that C-PTIO had no effect on the growth of cells. Therefore, C-PTIO was added externally on the sixth day for the next experiment.

C-PTIO was added at a final concentration of 80 μM, and its effect on the NO content was assessed 48 h after treatment in the experimental group and the control group. As shown in Figure 5A–C, during the incubation period of 48 h under the three culture conditions, the NO content in the C-PTIO treatment group was significantly reduced compared with the control group without C-PTIO. These results indicat that the exogenous addition of C-PTIO to the culture medium effectively abrogated the induction of NO by SA and JA in *N. flagelliforme*.

To assess the possible downstream effects of blocked NO signaling, the capsular polysaccharides in the C-PTIO treated group and the untreated group were determined. As shown in Figure 5D, the SA and JA induced cultures of *N. flagelliforme* without C-PTIO treatment showed a significant increase in CPS, by 30 and 28%, respectively, indicating that SA and JA had an obvious promoting effect on the accumulation of CPS. At the same time, the exogenous addition of C-PTIO reduced the content of NO and decreased the synthesis and secretion of CPS by the cells, which also indicated that NO plays an important intermediary role in the promotion of the synthesis and secretion of the CPS by SA and JA in *N. flagelliforme*.

#### 3.4.3. The Effect of the Exogenous NO Donor SNP on the NO Level and Polysaccharide Production

SNP is a commonly used NO donor [42]. To verify the causative relationship between the NO and the polysaccharide accumulation in *N. flagelliforme*, different concentrations of SNP were added exogenously, followed by an analysis of the biomass, NO content, and polysaccharide production.

Appendix A shows the effect of different concentrations of exogenous SNP on the biomass of *N. flagelliforme*, which had no significant change on the sixth day compared with the control group. This indicates that SNP did not affect the normal growth of *N. flagelliforme*, so exogenous SNP was added on the 6th day for the subsequent experiments.

As shown in Figure 6A, the NO content of *N. flagelliforme* increased with the increase in SNP concentration (125 and 250 μM) three days before treatment compared with the control group. In particular, after 12 h of SNP treatment, the NO content in the experimental groups with 125, 250, and 500 μM SNP was 10, 30, and 40% higher than that in the control group without SNP, respectively. It is speculated that SNP could increase the NO content in *N. flagelliforme*.

Compared with the control group without SNP treatment, the CPS of the experimental groups with different concentrations of SNP increased significantly, and the polysaccharide production of the experimental groups with 125, 250, and 500 μM SNP increased by 26%, 32%, and 28%, respectively, compared with the control group (Figure 6B). A previous study reported that NO acts as a signaling molecule that promotes the production of certain metabolites [39]. Therefore, the above results showed that extrinsically added NO released from SNP at different concentrations promoted polysaccharide synthesis in *N. flagelliforme*.

According to the analysis of the relationship between NO and CPS (Figure 7), CPS production was positively associated with the NO contents, and it is suggested that CPS production was correlated with the levels of metabolic level induced by the NO increased CPS accumulation. NO recently emerged as a key signaling molecule in plants. Studies have shown that NO acts as a signal molecule involved in the elicitor-induced defense responses of plants [19]. At the same time, the activation of the endogenous signaling pathways has been well documented to play key roles in regulating the accumulation of secondary metabolites in plants. NO has been well demonstrated to act as a signaling molecule that modifies cellular responses against different stressors in the corresponding signal transduction pathways [39]. Therefore, it can be speculated that SA and JA can be used as exogenous inducers to stimulate *N. flagelliforme* cells to produce NO through the corresponding metabolic pathways, and NO, in turn increases the production of CPS as a response signal.

## 4. Conclusions

In this study, chemical induction with SA and JA significantly promoted the accumulation of *N. flagelliforme* polysaccharides. The contents of some chemical components were selectively affected by the culture conditions, while the proportion of the monosaccharide composition, functional groups, and conformation of the polysaccharides were not greatly affected. The antioxidant results showed that all three polysaccharides exhibited a high antioxidant activity. At a low concentration, the control-CPS without chemical induction exhibited a strong scavenging activity, but at high concentration, the scavenging activity did not increase further. According to the results obtained with the external addition of an NO donor or a scavenger, SA and JA most likely induced the accumulation of polysaccharides by increasing the intracellular NO levels in *N. flagelliforme*.

These results demonstrate that proper chemical modulators can enhance the accumulation of polysaccharides without affecting their structure and activity. These results offer a new perspective for the study of the mechanisms driving the influence of plant hormones on polysaccharide synthesis in *N. flagelliforme* and lay a theoretical foundation for enhancing the yield of bioactive polysaccharides by regulating intracellular NO levels.

## Figures and Tables

**Figure 1 foods-12-00915-f001:**
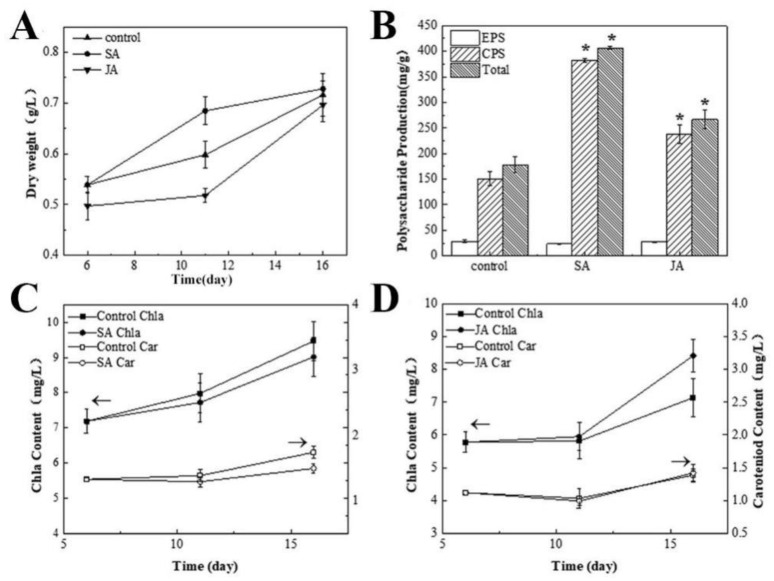
The effects of SA and JA on the biomass (**A**), polysaccharide production (**B**), and photosynthetic pigments (**C**,**D**) of *N. flagelliforme*. ** p* < 0.05, compared with the control group.

**Figure 2 foods-12-00915-f002:**
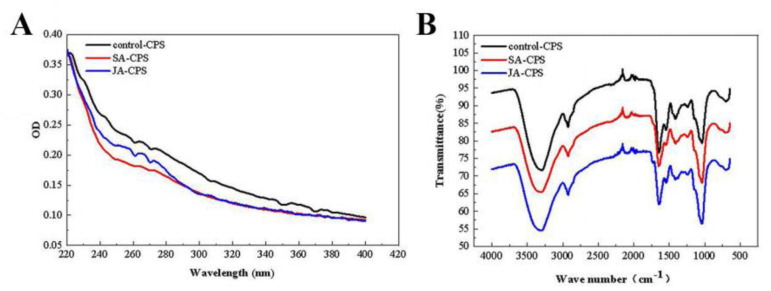
UV (**A**) and FT−IR (**B**) spectra of the control-CPS, SA-CPS, and JA-CPS.

**Figure 3 foods-12-00915-f003:**
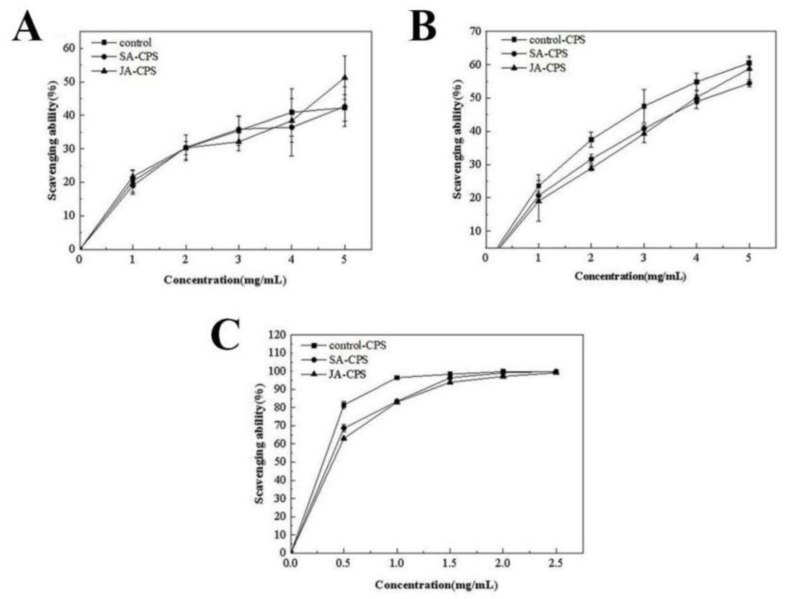
Antioxidant activities including the DPPH radical scavenging activity (**A**), the hydroxyl radical scavenging activity (**B**), and the ABTS^+^ radical scavenging activity (**C**) of the control-CPS, SA-CPS, and JA-CPS.

**Figure 4 foods-12-00915-f004:**
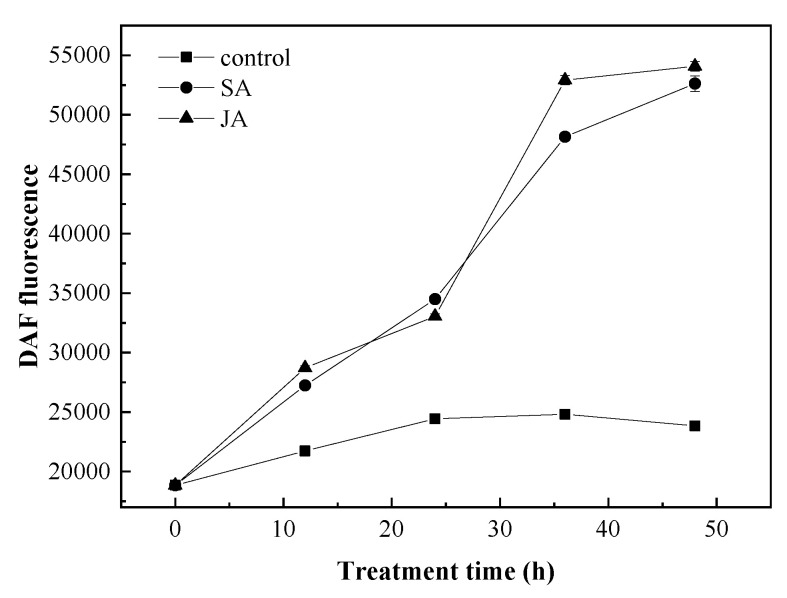
The NO content in *N. flagelliforme* under SA and JA induction.

**Figure 5 foods-12-00915-f005:**
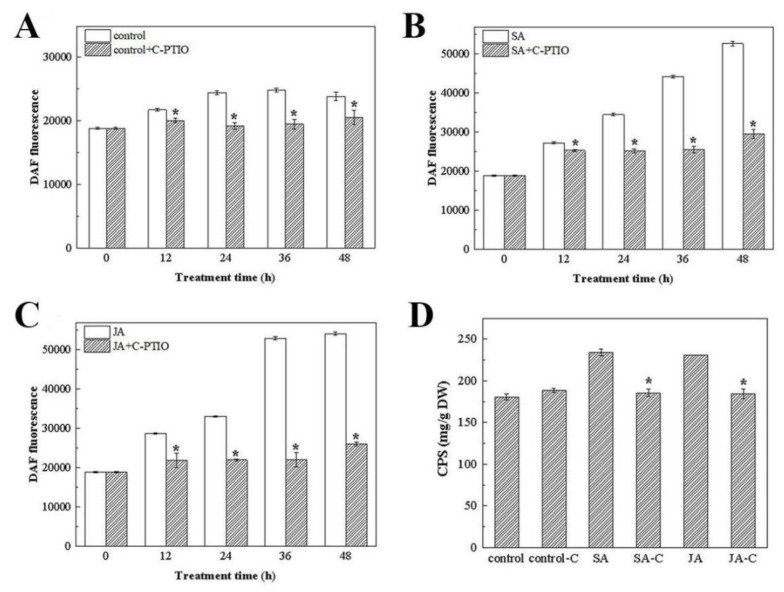
Effect of the C-PTIO on the NO content (**A**–**C**) and polysaccharide production (**D**) in *N. flagelliforme* under salicylic acid and jasmonic acid. (**A**) ** p* < 0.05, compared with the control group, (**B**) ** p* < 0.05, compared with the SA group, (**C**) ** p* < 0.05, compared with the JA group, and (**D**) ** p* < 0.05, compared with the SA and JA group.

**Figure 6 foods-12-00915-f006:**
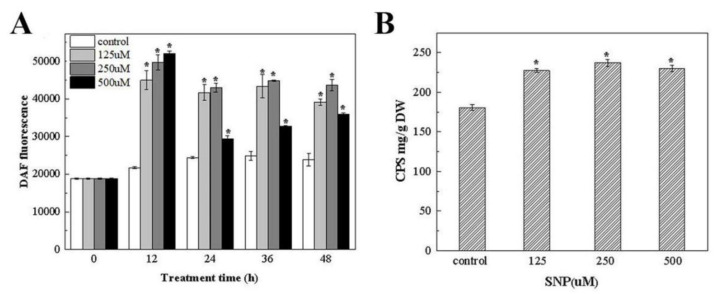
The effect of different SNP concentrations on the NO content (**A**) and the polysaccharide production (**B**) of *N. flagelliforme*. ** p* < 0.05, compared with the control group.

**Figure 7 foods-12-00915-f007:**
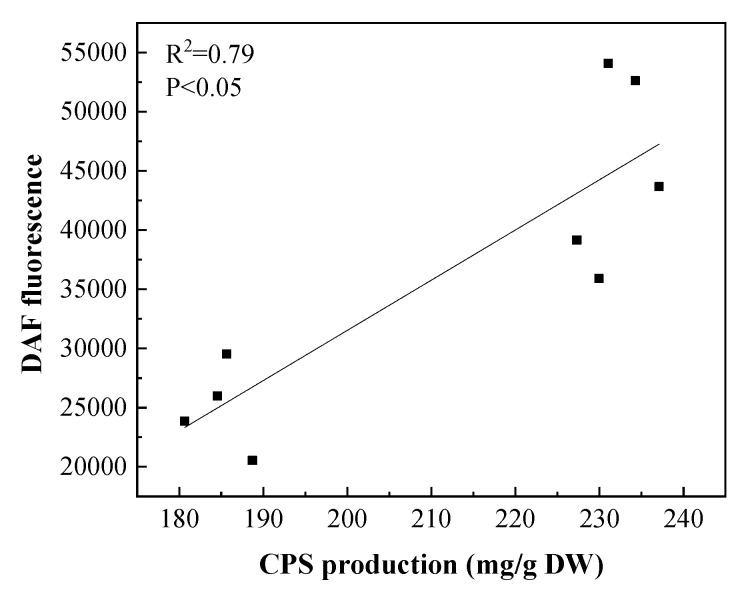
Relationship between changes in CPS production and NO under different culture conditions.

**Table 1 foods-12-00915-t001:** The effects of the chemical inducers on the polysaccharides production of *N. flagelliforme*.

Compound Classification	Compound	Optimum Concentration	Increase in Polysaccharide Production (%)
Phytohormones	2,4-D	0.25 mg/L	109.7
	GA	10 mg/L	30.19
Metal ions	Ca^2+^	10 mg/L	27.23
	Mg^2+^	-	-
Vitamins	VB_12_	-	-
Oxidizing agents	AAPH	-	-
	MB	3 mg/L	15.54
	H_2_O_2_	60 mg/L	18.92
Antioxidants	PG	200 mg/L	86.15
Signaling reagents	JA	1 mg/L	58.57
	SA	1 mg/L	35.33
	AHLs	0.5 mg/L	74.17

-, not significant.

**Table 2 foods-12-00915-t002:** The basic physicochemical properties of the control-CPS, SA-CPS, and JA-CPS.

Basic Physicochemical Properties		Sample Name		
		Control-CPS	SA-CPS	JA-CPS
Chemical composition (%)	Total sugar	84.47 ± 2.17	87.67 ± 3.23	89.59 ± 2.75
	Protein	0.49 ± 0.04	0.54 ± 0.02	0.44 ± 0.03
	Uronic acid	15.03 ± 1.32	11.79 ± 1.25	10.17 ± 1.46
	Total polyphenols	ND	ND	ND
Average molecular weights (kDa)		2.06 × 10^3^	2.16 × 10^3^	2.04 × 10^3^
Monosaccharide composition (percentage %)	Xylose	0.85 ± 0.01	0.82 ± 0.11	0.68 ± 0.01
	Arabinose	1.81 ± 0.01	2.08 ± 0.02	1.63 ± 0.04
	Ribose	2.98 ± 0.01	3.59 ± 0.02	3.47 ± 0.16
	Rhamnose	5.60 ± 0.12	6.67 ± 0.01	4.75 ± 0.11
	Fucose	1.98 ± 1.04	1.86 ± 0.03	1.28 ± 0.04
	Fructose	0.33 ± 0.05	0.44 ± 0.05	0.57 ± 0.02
	Mannose	16.87 ± 0.15	18.45 ± 0.26	16.41 ± 0.16
	Galactose	17.84 ± 0.18	19.74 ± 0.33	17.28 ± 0.12
	Glucose	46.39 ± 0.62	40.44 ± 0.44	48.80 ± 0.19
	Glucuronic acid	4.97 ± 0.16	5.89 ± 0.99	5.11 ± 0.36

ND, not detected.

## Data Availability

The data presented in this study are available upon request from the corresponding author.

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
