# Peer review of "Salicylic Acid and Jasmonic Acid Increase the Polysaccharide Production of *Nostoc flagelliforme* via the Regulation of the Intracellular NO Level"

_foods, 2023, doi:10.3390/foods12050915_

Round 1
Reviewer 1 Report
Cheng-Feng Han et al. described a very nice and innovative work. And even he has tried his best to described in a most easiest way to improve the synthesis of secondary metabolites.
The complete work is very concisely described besides need moderate English correction.
I am highly recommending the publication of this manuscript.
Author Response
Reviewer 1:
The complete work is very concisely described besides need moderate English correction.
Response: We appreciate it very much for this good suggestion, and we have done it according to your ideas. We used the English editorial services recommended by the journal to revise this manuscript.The specific modifications have been presented in the revised version, please check.

Reviewer 2 Report
The purpose of the study was to improve the polysaccharide production by Nostoc flagelliforme. The goal was also to analyze the influence of 12 chemicals on polysaccharide accumulation. The manuscript is an interesting piece of work and may be interested to the readers. The experiments were well designed and performed. The tables and figures were clearly presented. However it should be improved/revised. Additionally, extensive editing of English language and style are required. The authors should send the manuscript to the native speaker. My recommendation is to accept the article for the possible publication in “Foods” after the revision.
Introduction
1. The overview on the state of the art is safficient. However lines 74-79 describe the results of the study not the purpose. I suggest authors to think about what relevant goal or hypotesis they would like to provide in this paragraph instead of describing the results.
Materials and Methods
1. Authors should revise the methodology section. There is a need to describe the experiments clearly.
2. ,,Materials and methods sections“: The authors mentioned (Tianjin, China, Beijing, China, SHIMADZU Corporation, Japan, but they did not include companies, cities and countries (which should be in brackets) e.g.: Tianjin, China (company, city country).
Results and discussion
The results and discussion are written clearly.
Refernces
References should be improved throughout. The authors should revise this section according to publisher instructions.
Author Response
Reviewer 2:
The purpose of the study was to improve the polysaccharide production by Nostoc flagelliforme. The goal was also to analyze the influence of 12 chemicals on polysaccharide accumulation. The manuscript is an interesting piece of work and may be interested to the readers. The experiments were well designed and performed. The tables and figures were clearly presented. However it should be improved/revised. Additionally, extensive editing of English language and style are required. The authors should send the manuscript to the native speaker. My recommendation is to accept the article for the possible publication in “Foods” after the revision.
Introduction
- The overview on the state of the art is safficient. However lines 74-79 describe the results of the study not the purpose. I suggest authors to think about what relevant goal or hypotesis they would like to provide in this paragraph instead of describing the results.
Materials and Methods
- Authors should revise the methodology section. There is a need to describe the experiments clearly.
- ,,Materials and methods sections“: The authors mentioned (Tianjin, China, Beijing, China, SHIMADZU Corporation, Japan, but they did not include companies, cities and countries (which should be in brackets) e.g.: Tianjin, China (company, city country).
Results and discussion
The results and discussion are written clearly.
References
References should be improved throughout. The authors should revise this section according to publisher instructions.
Response:
1: We used the English editorial services recommended by the journal to revise this manuscript.
2: Rewrite the last paragraph of the introduction as: Here, we screened SA and JA among 12 chemical inducers that can significantly improve the polysaccharide production of N. flagelliforme cells. These two inducers not only improved the yield of polysaccharides but also maintained their original structure and function. In order to explore the possible mechanism of SA and JA inducing N. flagelliforme polysaccharide synthesis and provide a new idea for the accumulation of N. flagelliforme polysaccharide, we measured the content of NO in N. flagelliforme cells induced by SA and JA, and found that NO played a key role in the process of N. flagelliforme polysaccharide synthesis.
3: We have added some specific methods of indicators in the materials and methods.
The extraction and purification methods of CPS were described in lines 112 to 124.
The determination of antioxidant index was described after line 151.
4: We are very sorry for our incorrect writing and it is rectified at line 97 and 145.
5: References have been checked and corrected.
Reviewer 3 Report
The current manuscript reports that salicylic acid and jasmonic acid increase the polysaccharide production of Nostoc flagelliforme without influencing the physicochemical properties and antioxidant activities via regulation of intracellular NO level. In general, this is an important and interesting work, well-written, logically structured. I have, however a few comments or suggestions.
I think that the title of the article is too long, I recommend reformulating.
In the Introduction, in the last paragraph, it is necessary to formulate the aim of the research, and not to generalize the results obtained.
In section 2.3. Extraction and purification of the CPS, briefly describe the methodology and the equipment you use.
Section 2.5. Chemical analysis rewrite in one style
Section 2.9. Statistical analysis, specify the number of repetitions.
As I understand it, table 1 is based on data from other studies, please add the reference in the title of the table.
In section 3.3.2 check the given test results and units of measurement.
Author Response
Reviewer 3:
I think that the title of the article is too long, I recommend reformulating.
In the Introduction, in the last paragraph, it is necessary to formulate the aim of the research, and not to generalize the results obtained.
In section 2.3. Extraction and purification of the CPS, briefly describe the methodology and the equipment you use.
Section 2.5. Chemical analysis rewrite in one style
Section 2.9. Statistical analysis, specify the number of repetitions.
As I understand it, table 1 is based on data from other studies, please add the reference in the title of the table.
In section 3.3.2 check the given test results and units of measurement.
Response:
1: The title was restated as follows: Salicylic acid and jasmonic acid increase the polysaccharide production of Nostoc flagelliforme via the regulation of the intracellular NO level
2: Rewrote the introduction as follows: Here, we screened SA and JA among 12 chemical inducers that can significantly improve the polysaccharide production of N. flagelliforme cells. These two inducers not only improved the yield of polysaccharides but also maintained their original structure and function. In order to explore the possible mechanism of SA and JA inducing N. flagelliforme polysaccharide synthesis and provide a new idea for the accumulation of N. flagelliforme polysaccharide, we measured the content of NO in N. flagelliforme cells induced by SA and JA, and found that NO played a key role in the process of N. flagelliforme polysaccharide synthesis.
3:
2.3. Extraction and purification of the CPS
In this paper, the crude polysaccharides were extracted on the base of reference literature.
The cell culture solution was centrifuged at 4000 r for 10 min, the cells were collected, frozen and dried them. The freeze-dried cells were added distilled water and then bathed in water at 80℃ for 4 hours. The supernatant was collected by centrifugation. The content of capsular polysaccharide in the supernatant could be determined by phenol-sulfuric acid method.
After the supernatant was concentrated by ultrafiltration, 80% ethanol was added to precipitate it, and it was placed at 4℃ overnight, centrifuged, collected the precipitate, and freeze-dried to obtain the capsule polysaccharide powder. The capsule polysaccharide powder was prepared into polysaccharide aqueous solution, separated by DEAE-52 column chromatography, purified by Sephadex G-100 column chromatography, and impurities were removed. The collected sugar-containing liquid was freeze-dried to obtain the purified capsule polysaccharide of N. flagelliforme, and stored at -80℃ for standby.
4:
2.5. Chemical analysis
The total sugar was determined according to the phenol-sulfuric acid method. The total protein content was determined by Bradford's method using bovine serum albumin as the standard. For the determination of uronic acid content, please refer to the method of Fan Liuping et al. The total polyphenol contents were determined by Folin–Ciocalteu’s reagent according to the national standard method of the People's Republic of China. The monosaccharide compositions of polysaccharides were analyzed with reference to previous reports. The homogeneity and molecular weight (MW) of polysaccharides were analyzed according to the method previously reported.
5:
2.9. Statistical analysis
The data were analyzed by analysis of variance (ANOVA) or independent-samples t-test using the SPSS statistical software (version 20.0). All samples were run in triplicate. The significance level was set at p < 0.05.
6: The data in Table 1 was the basis of this experiment. SA and JA were selected by measuring the accumulation of 12 chemicals on the content of polysaccharides, and the next experiment was carried out. The dosage of 12 chemicals was obtained according to the literature and previous accumulation in the laboratory. The improvement rate of polysaccharides was not obtained from the references, but determined by specific experiments.
7: Rewrote some conclusions in section 3.3.2 as follows:
The vibration absorption peak at 1200 and 900 cm − 1 was the C-O-C bond stretching vibration peak of pyran ring in control-CPS, SA-CPS and JA-CPS.